EMBO
Molecular Medicine

# Lentiviral vectors escape innate sensing but trigger p53 in human hematopoietic stem and progenitor cells

Francesco Piras[1,2], Michela Riba[3], Carolina Petrillo[1,2], Dejan Lazarevic[3], Ivan Cuccovillo[1],
Sara Bartolaccini[1], Elia Stupka[3,†], Bernhard Gentner[1], Davide Cittaro[3], Luigi Naldini[1,2] &
Anna Kajaste-Rudnitski[1,*] (iD)

## Abstract

Clinical application of lentiviral vector (LV)-based hematopoietic stem and progenitor cells (HSPC) gene therapy is rapidly becoming a reality. Nevertheless, LV-mediated signaling and its potential functional consequences on HSPC biology remain poorly understood. We unravel here a remarkably limited impact of LV on the HSPC transcriptional landscape. LV escaped innate immune sensing that instead led to robust IFN responses upon transduction with a gamma-retroviral vector. However, reverse-transcribed LV DNA did trigger p53 signaling, activated also by non-integrating Adeno-associated vector, ultimately leading to lower cell recovery *ex vivo* and engraftment *in vivo*. These effects were more pronounced in the short-term repopulating cells while long-term HSC frequencies remained unaffected. Blocking LV-induced signaling partially rescued both apoptosis and engraftment, highlighting a novel strategy to further dampen the impact of *ex vivo* gene transfer on HSPC. Overall, our results shed light on viral vector sensing in HSPC and provide critical insight for the development of more stealth gene therapy strategies.

**Keywords** gene therapy; hematopoietic stem and progenitor cells; innate sensing; lentiviral vectors; p53 signaling
**Subject Categories** Genetics, Gene Therapy & Genetic Disease; Haematology

## Introduction

Encouraging clinical results of recent hematopoietic stem cell (HSC) gene therapy trials are opening novel perspectives for the treatment of monogenic diseases affecting the hematopoietic system. Current gene transfer protocols are based on the modification of bone marrow (BM) or mobilized peripheral blood (mPB)-derived CD34[+] cells, a cell population enriched in HSC but containing also a large fraction of more committed progenitor cells, termed all together hematopoietic stem and progenitor cells (HSPC). In this setting, self-inactivating (SIN) lentiviral vectors (LV) have been successfully used in clinical trials for the treatment of several diseases, as reviewed in Naldini (2015). Nevertheless, multiple incubations with high vector doses and prolonged *ex vivo* culture are required to reach clinically relevant transduction levels, potentially impacting HSPC biological properties (Kajaste-Rudnitski & Naldini, 2015).

Lentiviral vectors rely on the same cellular machinery as HIV-1 to reach the nuclear compartment of target cells and integrate within the host genome. During these steps, LV nucleic acids and proteins can potentially be recognized by innate sensors. HIV genomic RNA can activate the cytosolic RNA sensor RIG-I (Berg *et al*, 2012), and cytosolic HIV DNA generated during reverse-transcription can be sensed by cGAMP synthase (cGAS) in specific cellular contexts, ultimately leading to IFN signaling (Gao *et al*, 2013). Other cytosolic nucleic acid sensors such as the DNA-dependent activator of interferon-regulatory factors (DAI) could also contribute to sensing of viral replication intermediates (Takaoka *et al*, 2007). In the context of LV-based gene therapy, innate sensing could be exacerbated by the absence of viral accessory proteins known to help immune escape of infectious HIV-1, envelope pseudotypes that impact the route of entry, the high vector doses that are significantly superior to those of a typical initial infection by HIV-1, and the presence in the vector stocks of DNA or other contaminants carried over from the producer cells (Kajaste-Rudnitski & Naldini, 2015). Indeed, LV have been shown to trigger TLR-dependent innate signaling in immune cells and hepatocytes (Rossetti *et al*, 2011; Agudo *et al*, 2012). In HSC, innate immune cues are emerging as regulators of stem cell functions and hematopoietic output (King & Goodell, 2011) at the expense of HSC self-renewal and/or maintenance (Essers *et al*, 2009; Baldridge *et al*, 2010) and accumulation of DNA damage and senescence (Walter *et al*, 2015; Yu *et al*, 2015). Only a

1 San Raffaele Telethon Institute for Gene Therapy, IRCCS San Raffaele Scientific Institute, Milan, Italy
2 Vita-Salute San Raffaele University, Milan, Italy
3 Center for Translational Genomics and Bioinformatics, IRCCS San Raffaele Scientific Institute, Milan, Italy
*Corresponding author. Tel: +39 0226435007; E-mail: kajaste.anna@hsr.it
†Present adress: Boehringer Ingelheim, Biberach an der Riß, Germany

 

handful of studies have addressed the impact of innate immune signaling in human HSPC indicating that TLR triggering can lead to skewed differentiation and cause apoptosis (Sioud *et al*, 2006; De Luca *et al*, 2009; Guo *et al*, 2010; Liu *et al*, 2012).

HIV integration into the host genome has also been shown to activate DNA damage responses (DDR) in the U2OS osteosarcoma cell line as well in primary CD4[+] T cells (Lau *et al*, 2005; Cooper *et al*, 2013).The p53 tumor suppressor plays critical roles in several DDR pathways and has been shown to be essential for maintaining quiescence during steady-state hematopoiesis in murine HSPC (Liu *et al*, 2009). Lower levels of p53 have been shown to confer a competitive advantage over HSC expressing more p53 (Bondar & Medzhitov, 2010) and to rescue human HSPC from apoptosis immediately after low-level irradiation (Milyavsky *et al*, 2010).

On these premises, exposure of human HSPC to LV during *ex vivo* gene transfer could trigger signaling mimicking host cell responses to viral infection with potential short- and long-term consequences that have not been addressed to date. We have investigated here how lentiviral transduction alters the global transcriptional landscape of human HSPC, impacting on their biological properties, shed light on the molecular mechanisms involved, and provide proof-of-principle on how to dampen these effects in the context of *ex vivo* gene therapy.

# Results

### Lentiviral reverse-transcribed DNA triggers p53 signaling independently of integration in human HSPC

We performed a time-course RNA-Seq analysis on cord blood (CB)-derived CD34[+] cells pre-stimulated with early-acting cytokines for 24 h and then exposed to either research- or clinical-grade VSV-g

pseudotyped (SIN) LV at a high multiplicity of infection, matching current clinical vector dose requirements. As controls, cells were exposed to Poly(I:C), non-infectious LV particles lacking the VSV-g envelope (Bald) or heat-inactivated vectors to control for contaminants co-administered with the LV or kept in culture untreated (Fig 1A). The greatest transcriptional variance within our dataset was time in culture, as samples clustered in three distinct temporal groups following principal component analysis (PCA), independently of the treatment group (Fig EV1A). The mere culture of HSPC in the presence of growth-promoting cytokines resulted in the transcriptional modulation of around 6,000–9,000 genes over time for all treatment categories (Appendix Table S1). For untreated HSPC, the most enriched pathway was the MAPK signaling (Fig EV1B), in accordance with growth factor and cytokine-induced stimulation (Geest & Coffer, 2009). Poly(I:C)-exposed HSPC strongly up-regulated innate immune responses, significantly mobilizing a total of 2691 genes (nominal $P < 0.05$) as compared to untreated controls (Appendix Table S1). We performed term enrichment analysis of the significantly modulated genes to highlight the biological processes affected by Poly(I:C) exposure in HSPC. Regulation of immune responses, NF-κB signaling, antiviral responses, programmed cell death, and antigen processing were among the most represented categories (Fig EV1B and data not shown). Remarkably, LV-exposed HSPC showed much milder responses, modulating 645 and 397 genes (nominal $P < 0.05$) with research- and clinical-grade vectors, respectively, when comparing to untreated cultured cells (Appendix Table S1). In a more stringent comparison with the respective transduction controls (Bald and inactivated purified LV), the number of genes modulated upon transduction further decreased to 321 and 281 genes (nominal $P < 0.05$) with research- and clinical-grade vectors, respectively (Appendix Table S1), converging significantly into the DDR and in particular the p53 signaling pathway (Fig 1B). No evidence of significant innate immune activation related to TLR-signaling or

---

**Figure 1.  LV transduction induces p53 signaling in human HSPC.**

A   Scheme of the RNA-Seq experiment performed on human cord blood (CB)-derived CD34[+] cells exposed to laboratory-grade (Lab LV) or purified (purified LV) LV at a multiplicity of infection (MOI) of 100 or p24 equivalent of non-transducing Env-less (Bald LV), heat-inactivated purified LV, or Poly(I:C).

B   Pathway enrichment analysis of the differentially expressed genes over time ($P ≤ 0.05$) for Bald LV vs. Lab LV and inactivated purified LV vs. the purified LV.

C   Heatmap showing the profile over time of the most significantly ($P ≤ 0.01$) differentially expressed genes for Lab LV vs. Bald LV.

D   Genes differentially expressed for Lab LV vs. Bald LV are boxed in red in the KEGG p53 signaling pathway.

E   Gene expression levels of p53-induced genes in CB-CD34[+] cells 48 h after LV or Env-less (Bald) LV exposure at MOI 100 or p24 equivalent (mean ± SEM, $n = 8$ for p48, PUMA and PHLDA3, $n = 21$ for p21, Wilcoxon signed-rank test, **$P = 0.0078$ for p48 PUMA and PHLDA3, ****$P < 0.0001$ for p21.

F   CB-CD34[+] were sorted for the fractions enriched in stem cells (CD133[+]CD38[−]) and the more committed progenitors (CD133[+]CD38[int] and CD133[+]CD38[hi]).

G, H   p21 mRNA levels were measured in the sorted fractions and bulk CB/BM-CD34[+] 48h after transduction at MOI 100 or p24 equivalent of controls (mean ± SEM, $n = 5$ for 133[+]38[−], $n = 8$ for 133[+]38[int] and 133[+]38[hi], Dunn's adjusted Kruskal–Wallis test, *$P ≤ 0.05$, **$P ≤ 0.01$, G), (mean ± SEM, in CB $n = 17$ for Bald and LV, $n = 11$ for LV and $n = 9$ for Empty LV, in BM $n = 8$ for Bald and LV, $n = 5$ for IDLV and Empty LV, Dunn's adjusted Kruskal–Wallis test, **$P ≤ 0.01$, ***$P ≤ 0.001$, H).

I   p21 mRNA levels measured in CB-CD34[+] transduced in the presence of the integrase inhibitor Raltegravir (Ral) or reverse-transcriptase inhibitor Lamivudine (3TC) (mean ± SEM, $n = 6$ for Mock, $n = 7$ for DMSO and $n = 8$ for Raltegravir, $n = 7$ for 3TC, Dunn's adjusted Kruskal–Wallis test, *$P ≤ 0.05$, **$P ≤ 0.01$, ***$P ≤ 0.001$).

J   Western blot (left panel) and quantifications (right panel) were performed on whole mPB-CD34[+] cells extracts at 24 h post-transduction (mean ± SEM, $n = 4$ for p53Ser15, $n = 6$ for p53, $n = 5$ for p21). The quantification was on WB performed from both mPB-CD34[+] and CB-CD34[+]. One representative of four Western blot gels is shown.

K   p21 protein up-regulation evaluated by FACS 48 h after transduction (mean ± SEM, $n = 7$ for Mock, $n = 11$ for Bald, $n = 4$ for IDLV and $n = 9$ for LV, Dunn's adjusted Kruskal–Wallis test, **$P ≤ 0.01$, ***$P ≤ 0.001$).

L–N   p21 and ISG levels were measured in CB-CD34[+] exposed to LV, LV lacking the central polypurine tract (ΔcPPT) or γ-RV at MOI 100 or p24 equivalent of Bald control vector, or MOI 10,000 of Adeno-associated vector (AAV6) (mean ± SEM, $n = 20$ for Bald, $n = 18$ for LV, $n = 16$ for RV, and $n = 14$ for AAV6, Dunn's adjusted Kruskal–Wallis test, ***$P ≤ 0.001$, L), (mean ± SEM, $n = 14$, Dunn's adjusted Friedman test, *$P ≤ 0.05$, ***$P ≤ 0.001$, Wilcoxon signed rank test, +++$P = 0.0004$, M), (mean ± SEM, for IRF7 and OAS1 $n = 18$ for Bald and LV, $n = 12$ for ΔcPPT and RV and $n = 8$ for AAV6, for ISG15 $n = 19$ for Bald, LV and ΔcPPT, $n = 8$ for RV and $n = 2$ for AAV6, Dunn's adjusted Kruskal–Wallis test, ***$P ≤ 0.001$, N).

Source data are available online for this figure.

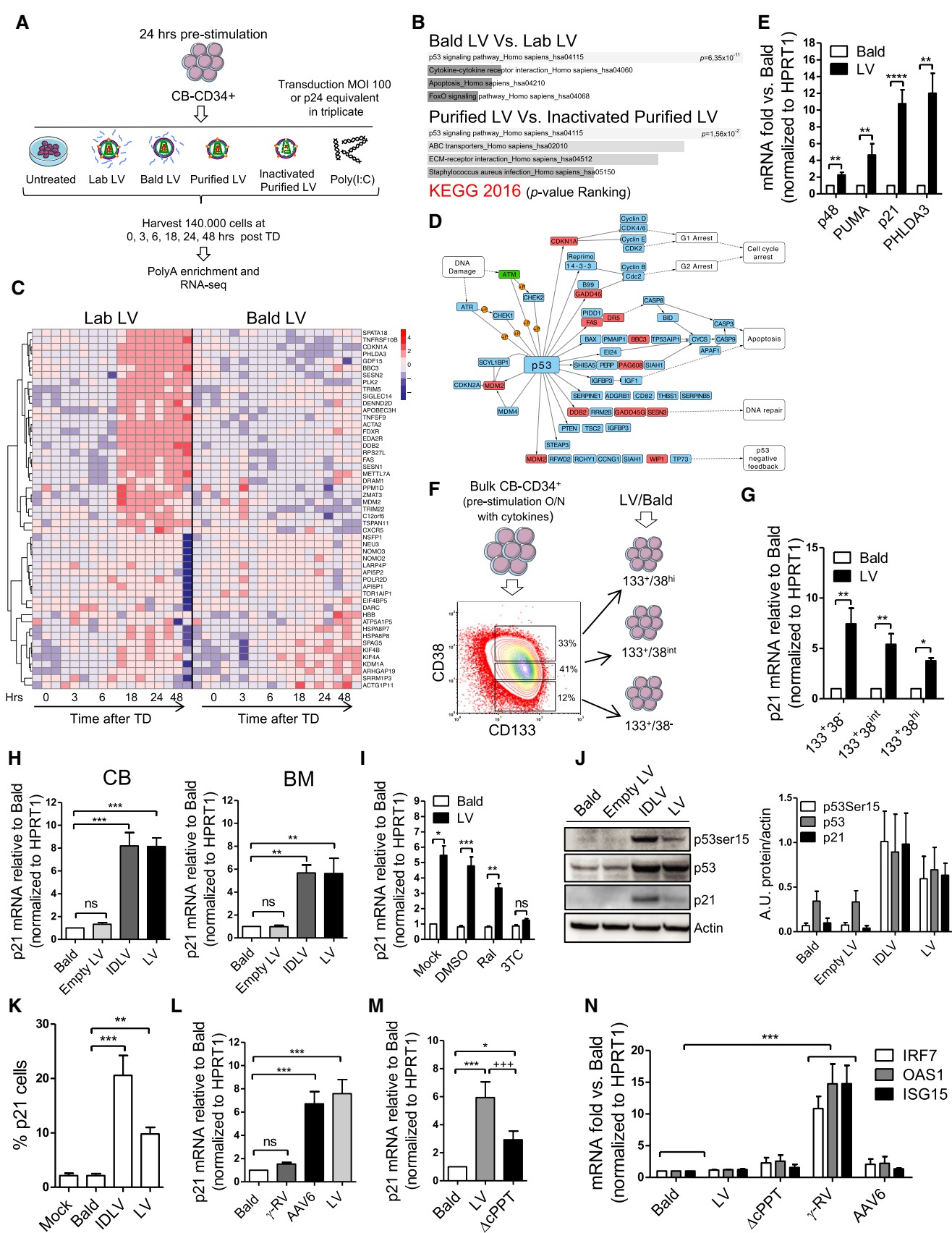

**Figure 1.**

activation of NF-kB/interferon-stimulated gene (ISG) transcription could be detected (Appendix Table S2). Instead, analysis of the differentially expressed genes with a nominal $P < 0.01$ in the LV-exposed HSPC compared to Bald-exposed controls revealed the up-regulation, around 18 h post-transduction, of a cluster of genes mapping to the KEGG p53 signaling pathway (Fig 1C and D). Up-regulation of some of the most significantly modulated genes involved in p53 signaling was confirmed by Taqman (Fig 1E) and using a clinical-grade LV (Fig EV1C).

To further investigate how LV induces p53 signaling in HSPC, we tracked p21 mRNA induction as a marker of p53 activation. Induction of p21 was dose-dependent and transient as levels normalized between control and transduced cells by day 5 in culture (Fig EV1D and E). It was also specific to human HSPC as CD4$^+$ T cells, Lin$^-$ murine HSPC, and different human cell lines of hematopoietic origin did not up-regulate p21 upon LV exposure (Fig EV1F). Overall, LV-induced p21 expression to a similar extent in all CD34$^+$ subpopulations, although the most primitive CD34$^+$ CD133$^+$CD38$^-$ fraction showed higher levels of induction, possibly correlating with the higher transduction levels reached in this cell fraction (Figs 1F and G, and EV1G). Both integration-competent and integration-defective LV (IDLV) led to a similar induction of p21 in CB as well as in bone marrow (BM)-derived CD34$^+$ HSPC (Fig 1H) at comparable vector DNA input (Fig EV1H). Of note, triggering was not due to vector stock contaminants, LV particle entry alone nor exposure of cells to viral cores lacking the genome, as neither the Bald vector nor the genome-less Empty LV lead to up-regulation of p21 (Fig 1H). In agreement with the IDLV-mediated up-regulation, p21 induction still occurred in HSPC transduced in the presence of the integrase inhibitor Raltegravir (Fig 1I), despite an efficient block in LV integration (Fig EV1I). Integration-independent activation of p53 signaling was confirmed by Western blot, FACS and indirect immunofluorescence (IFI) staining in terms of phosphorylation of p53 at Serine 15, increase in basal p53 levels as well as induction of p21 by both LV and IDLV also in mobilized peripheral blood (mPB)-derived CD34$^+$ HSPC (Figs 1J and K, and EV1J–L). Furthermore, no changes in phosphorylated histone 2AX (γH2AX) foci were observed, in line with a DNA break-independent induction of p53 (Fig EV1L). Finally, p53 signaling was abrogated in the presence of the reverse-transcriptase inhibitor 3TC (Figs 1I and EV1I and J), suggesting that lentiviral DNA synthesis is required for p53 signaling to occur in HSPC.

Taken together, these observations indicate that reverse-transcribed unintegrated LV DNA specifically triggers p53 rather than innate immune signaling in human HSPC, despite cells can rapidly up-regulate these pathways upon Poly(I:C) exposure.

### p53 induction requires nuclear import of vector DNA

Besides LV, the other viral vectors most often used to introduce genetic material into cells are integrating gamma-retroviral vectors (γRV) and DNA-based non-integrating Adeno-associated vectors (AAV). We tested the capacity of both γRV and AAV-6, that efficiently transduces human HSPC (Wang *et al*, 2015), to induce p21 in HSPC. AAV-6-exposed HSPC showed robust p21 induction similar to LV, while VSV-g pseudotyped γRV only weakly affected p21 expression at routinely used vector doses (Figs 1L and EV1M and N), potentially due to the lower transduction efficiency of γRV and given the strong linear correlation between VCN and p21 responses (Fig EV1D and N). Both HIV-1-derived LV and AAV-6 actively import the vector DNA into the nucleus of target cells (Bushman *et al*, 2005; Nonnenmacher & Weber, 2012). To test whether nuclear import of vector DNA is involved in p53 signaling in HSPC in the context of LV, we generated a vector devoid of the central polypurine tract (cPPT) required for efficient nuclear import of the pre-integration complex (Follenzi *et al*, 2000; Zennou *et al*, 2000). Up to three-fold lower nuclear import of the ΔcPPT LV compared to unmodified LV was verified in both 293T cells and HSPC (Fig EV1O and P), and the former vector induced twofold less p21 mRNA compared to control LV in CD34$^+$ cells (Fig 1M), suggesting that efficient nuclear import of vector DNA is required to activate p53 signaling in HSPC.

As p53 signaling has recently been linked with type I IFN pathways (Hartlova *et al*, 2015; Yu *et al*, 2015), we also examined expression of several ISG in human HSPC after exposure to the different vectors. In agreement with our RNA-Seq dataset, no activation of ISG could be evidenced in human HSPC upon LV or AAV-6 transduction (Fig 1N). Conversely, γRV triggered significant up-regulation of ISG independently of reverse-transcription and integration, as ISG induction occurred also in the presence of the reverse-transcriptase inhibitor azidothymidine (AZT) or Raltegravir (Figs 1N and EV1Q). γRV-mediated ISG induction was inhibited when the type I IFN receptor signaling was blocked (Fig EV1Q), but it does not explain lack of p53 signaling upon γRV exposure, as pre-treatment of HSPC with type I IFN did not prevent induction of p21 by LV (Fig EV1R). Noteworthy, both LV and γRV readily triggered ISG expression in murine Lin$^-$ cells (Fig EV1S), indicating that the capacity of LV to avoid type I IFN activation is specific to human cells.

### Functional consequences of LV-induced signaling in HSPC

p53 has a pivotal role in regulating HSC quiescence and homeostasis during both steady-state hematopoiesis and under replicative stress

**Figure 2. Functional consequences of LV-mediated signaling in human HSPC *in vitro*.**

A    Cell proliferation of CB-CD34$^+$ cells transduced with MOI 100 of LV, p21 overexpressing LV (p21) or p24 equivalent of Bald (mean ± SEM, $n = 8$, Wilcoxon signed-rank test, *$P = 0.0156$, **$P = 0.0078$ in CD34$^+$133$^-$, CD34$^+$133$^+$, CD34$^+$133$^+$90$^+$, *$P = 0.0234$, **$P = 0.0078$ in Bulk CD34$^+$).

B    Cell cycle analysis 2 days after the transduction (mean ± SEM, $n = 6$).

C–E    Apoptosis was evaluated 2 days after exposure of total or sorted CB-CD34$^+$ to LV, integrase-defective LV (IDLV) or Env-less (Bald) at MOI 100 or p24 equivalents (mean ± SEM, $n = 12$, Dunn's adjusted Kruskal–Wallis test, *$P ≤ 0.05$, **$P ≤ 0.01$, ***$P ≤ 0.001$, C) (mean ± SEM, $n = 4$ for 133$^+$38$^-$, $n = 7$ for 133$^+$38$^{int}$ and 133$^+$38$^{hi}$, Wilcoxon signed-rank test, *$P = 0.0223$ in late apoptotic, *$P = 0.0156$ in live and early apoptotic, E). (D) CB-CD34$^+$ cells were counted 2, 4, and 6 days after the transduction (mean ± SEM, $n = 4$, Wilcoxon signed-rank test, **$P = 0.0039$).

F    Colony-forming unit (CFU) output after 2 weeks of differentiation (mean ± SEM, $n = 10$, Wilcoxon signed-rank test, *$P = 0.0170$).

G    Impact of integrase (Ral) or reverse-transcriptase (3TC) inhibition on apoptosis (mean ± SEM, $n = 8$ for Mock and Ral, $n = 7$ for DMSO and 3TC, Wilcoxon signed-rank test, *$P = 0.0156$, **$P = 0.0078$).

(Liu *et al*, 2009; Lane & Scadden, 2010; Milyavsky *et al*, 2010; Mohrin *et al*, 2010). The p21 induction observed upon LV transduction (Fig EV2A) led to a slight but significant delay in cellular proliferation rates, in particular in the most primitive $CD34^+CD133^+CD90^+$ fraction (Figs 2A and EV2B). Similar but sustained growth delay was observed in p21 overexpressing HSPC

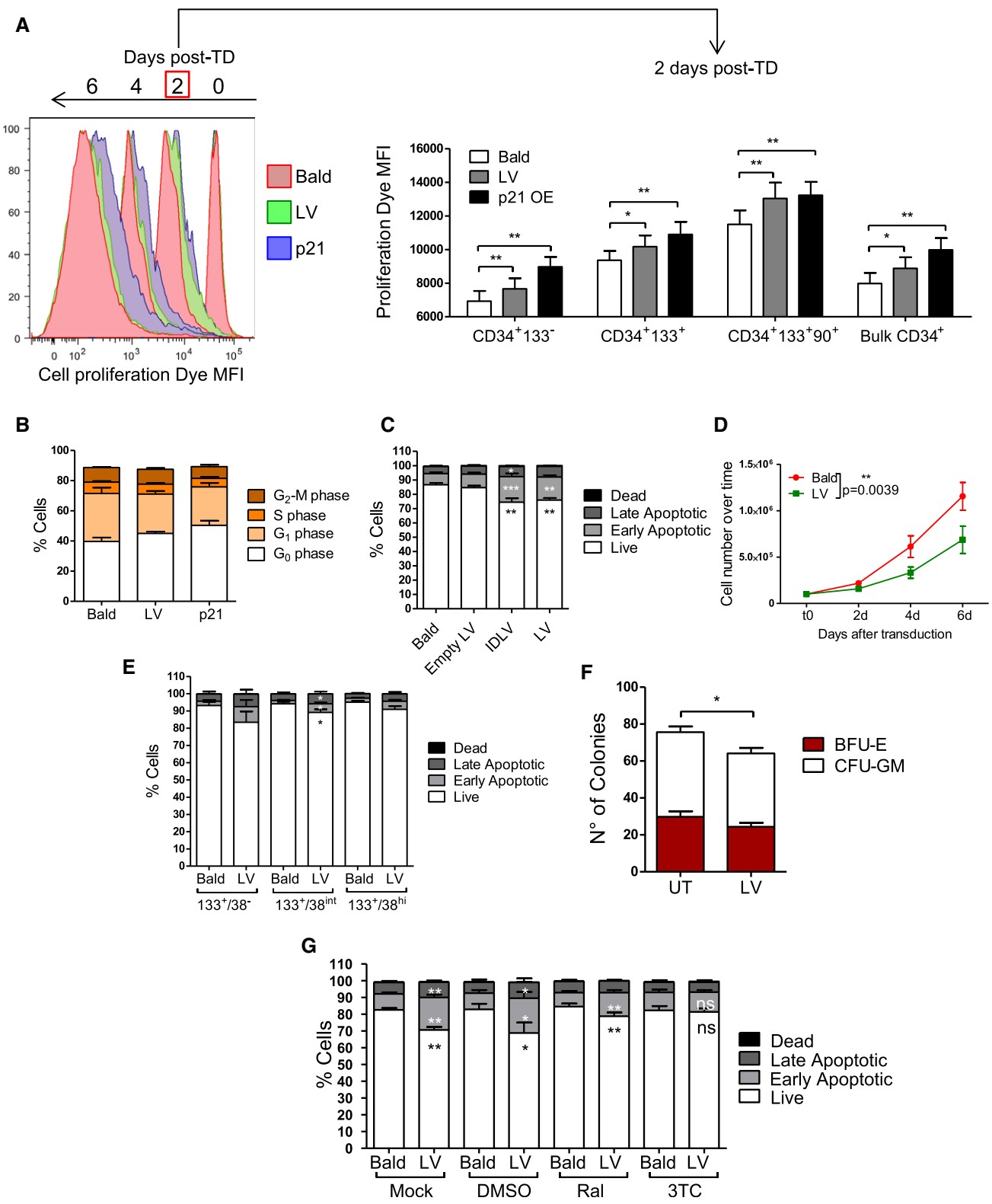

Figure 2.

used as a positive control for this assay (Figs 2A and EV2A and B). In agreement, the proportion of cells that had undergone fewer divisions (group C1 in Fig EV2C) tended to be more represented in the LV-exposed condition as compared to control cells in the bulk CD34$^+$ HSPC (Fig EV2D) and transduced HSPC displayed a higher fraction of cells in the $G_0$ cell cycle phase as compared to controls (Fig 2B). Furthermore, within the transduced population, a stronger proliferation delay was observed for the GFP$^{high}$ fraction, likely harboring more vector copies, compared to the GFP$^{low}$ ones (Fig EV2D). In agreement, higher p21 induction was detected by FACS in the more transduced BFP$^{high}$ cells (Fig EV1J).

LV-exposed HSPC showed a slight but significant dose-dependent increase in apoptotic cells (Figs 2C and EV2E). The combined effects of lower proliferation and apoptosis of LV-transduced HSPC were reflected also in terms of total cell counts over time in culture (Fig 2D). Apoptosis occurred to similar extent in all CD34$^+$ subpopulations (Fig 2E), and lower CFU-GM colony output was observed in LV-exposed total HSPC (Fig 2F) with the CD133$^+$CD38$^-$ and CD133$^+$CD38$^{int}$ fractions showing significantly lower CFU-GM counts as compared to controls (Fig EV2F), correlating with higher p53 activation (Fig 1G). Increased apoptosis followed also exposure of HSPC to AAV-6 and clinical-grade LV. Other p53-independent mechanisms may also lead to apoptosis in human HSPC, as also γRV-exposed cells showed similar percentages of apoptotic cells (Fig EV2G and H). Inhibition of reverse-transcription, but not of integration, completely inhibited LV-mediated induction of apoptosis in HSPC, correlating with their impact on p21 induction (Figs 1I and 2G).

To further determine how LV transduction affects human HSC function, we transplanted immunocompromised NSG mice with equal and limiting numbers of CB-derived CD34$^+$ HSPC exposed to either a transducing LV or a genome-less "Empty LV" control vector. To investigate an eventual selective disadvantage of transduced HSPC, a group of mice received a mix of LV and Empty LV-exposed cells in a 3:1 ratio (Fig 3A). The level of transduction, p53 activation, and consequent impact on cell viability and clonogenicity of transplanted HSPC was verified in vitro to be comparable to our prior results (Fig EV3A–E). Despite equal cellular input, LV-exposed HSPC showed a significantly lower engraftment at all timepoints compared to controls (Fig 3B). Decreased engraftment was confirmed also in mPB-derived HSPC transduced according to the current clinical standard protocol based on two subsequent rounds of transduction with a VSV-g pseudotyped clinical-grade LV (Figs 3C

and EV3F). Of note, no significant differences in the numbers of CD34$^+$ cells retrieved from the bone marrow of NSG mice 16 h after transplantation could be detected between transduced and control HSPC (Fig 3D), suggesting that LV transduction does not alter HSPC homing capacity. Once engrafted, no selective disadvantage of transduced HSPC over controls could be seen in the mix condition. Accordingly, the percentages of transduced GFP$^+$ cells remained constant over time (Fig EV3G). LV transduction did not alter lineage composition of human cells in periphery (Fig EV3H), but analysis of the bone marrow at 12 weeks post-transplantation reflected the levels of human cells in the peripheral blood and confirmed significantly lower engraftment of LV-exposed HSPC (Fig 3B). Nevertheless, no significant differences in the percentages of CD34$^+$ cells could be observed between the different groups, and equal frequency of more primitive CD34$^+$CD38$^-$ and committed CD34$^+$ CD38$^+$ cells was seen in the bone marrow (Fig 3E). Within the more primitive CD34$^+$CD38$^-$ fraction, the proportion of HSC, immature lymphoid progenitors (MLP) and multipotent progenitors (MPP; Doulatov et al, 2012; Notta et al, 2016) was also conserved between groups (Fig 3F) and no differences in lineage composition could be observed in the spleen of primary recipients (data not shown). Upon secondary transplantation, all mice were successfully repopulated independently of the treatment group (Fig 3G). Although a trend toward lower engraftment of LV-exposed HSPC was still observed in secondary recipients, the differences compared to the control group were no longer significant. Absence of selective advantage of untransduced HSPC was confirmed in this setting as the level of engraftment and percentage of GFP$^+$ cells remained stable over time in the mix condition (Figs 3G and EV3I). At the end of the experiment, no major differences in the bone marrow composition were seen in terms of total CD34$^+$ HSPC and frequency of more primitive CD34$^+$CD38$^-$ and committed CD34$^+$ CD38$^+$ cells (Figs 3H and EV3J). In agreement, a limiting dilution assay (LDA) further confirmed that LV transduction does not significantly alter the long-term repopulating stem cell frequencies (Fig 3I), although engraftment levels were again slightly lower in the LV-exposed group due to lower numbers of viable cells infused from matched treatment doses (Fig EV3K).

Overall, these results indicate that although exposure to LV negatively impacts HSPC maintenance ex vivo and engraftment in vivo due to acute induction of apoptosis, it does not affect their homing, composition, lineage output, or long-term repopulating capacity.

---

Figure 3. *In vivo* impact of LV-mediated signaling.

A    Schematic representation of the experimental design.

B    Percentages of total human CD45$^+$ cells were monitored in the peripheral blood over time and in the bone marrow at the end of the experiment (each dot representing a single mouse, $n$ = 10 for Empty LV, $n$ = 9 for Mix and LV, Dunn's adjusted Kruskal–Wallis test, *$P$ ≤ 0.05, **$P$ ≤ 0.01, ***$P$ ≤ 0.001).

C    Schematic representation of the experimental design and percentages of total human CD45$^+$ cells monitored in the peripheral blood over time and in the bone marrow at the end of the experiments (each dot representing a single mouse, $n$ = 5 for inact LV, $n$ = 4 for LV, Mann–Whitney test, *$P$ = 0.0491).

D    Percentages and absolute numbers of human CD34$^+$ cells retrieved from the bone marrow of NSG mice 16 h post-transplantation (each dot representing a single mouse, $n$ = 2 for not transplanted (no TP ctrl) mice, $n$ = 4 for Bald and LV).

E, F    Representative FACS plots and frequencies of CD34$^+$, CD34$^+$38$^+$, and CD34$^+$CD38$^-$ within the human CD45$^+$ cells and of CD45RA$^-$90$^+$ (HSC), CD45RA$^-$90$^-$ (MPP), and CD45RA$^+$90$^-$ (MLP) within the human CD34$^+$38$^-$ cells retrieved from the bone marrow of primary recipients (mean ± SEM, lower panel, or each dot representing a single mouse, upper right panel, $n$ = 10 for Empty LV, $n$ = 9 for Mix and LV, E), (mean ± SEM, $n$ = 7 for Empty LV and Mix, $n$ = 6 for LV, F).

G, H    Percentages of CD45$^+$ cells in the peripheral blood and frequency of CD34$^+$ cells within the bone marrow of the secondary recipients are shown (each dot representing a single mouse, $n$ = 10 for Empty LV, $n$ = 9 for LV and $n$ = 11 for Mix, G and H).

I    Table and plot of the calculated HSC frequency obtained by LDA for untransduced (Bald) and MOI 100 transduced (LV) CB-CD34$^+$ cells ($n$ = 2 in Bald and $n$ = 4 in LV for 3,000 cells, $n$ = 3 for 10,000 and 30,000 cells, $n$ = 4 for 90,000 cells, $n$ = 2 for not transplanted controls). Each dot represents a single mouse.

HSC = hematopoietic stem cells, MPP = multipotent progenitors, MLP = multilymphoid progenitors.

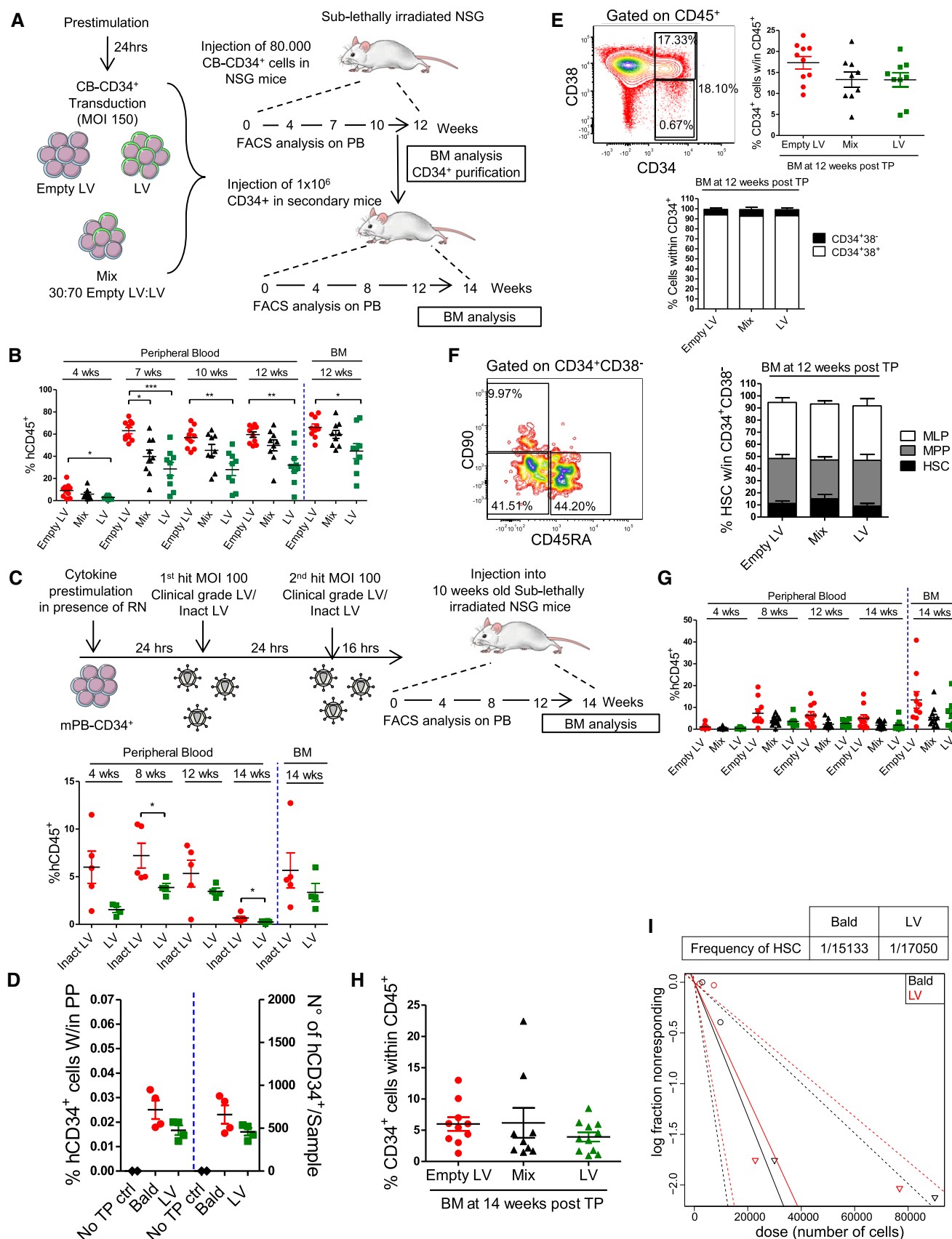

**Figure 3.**

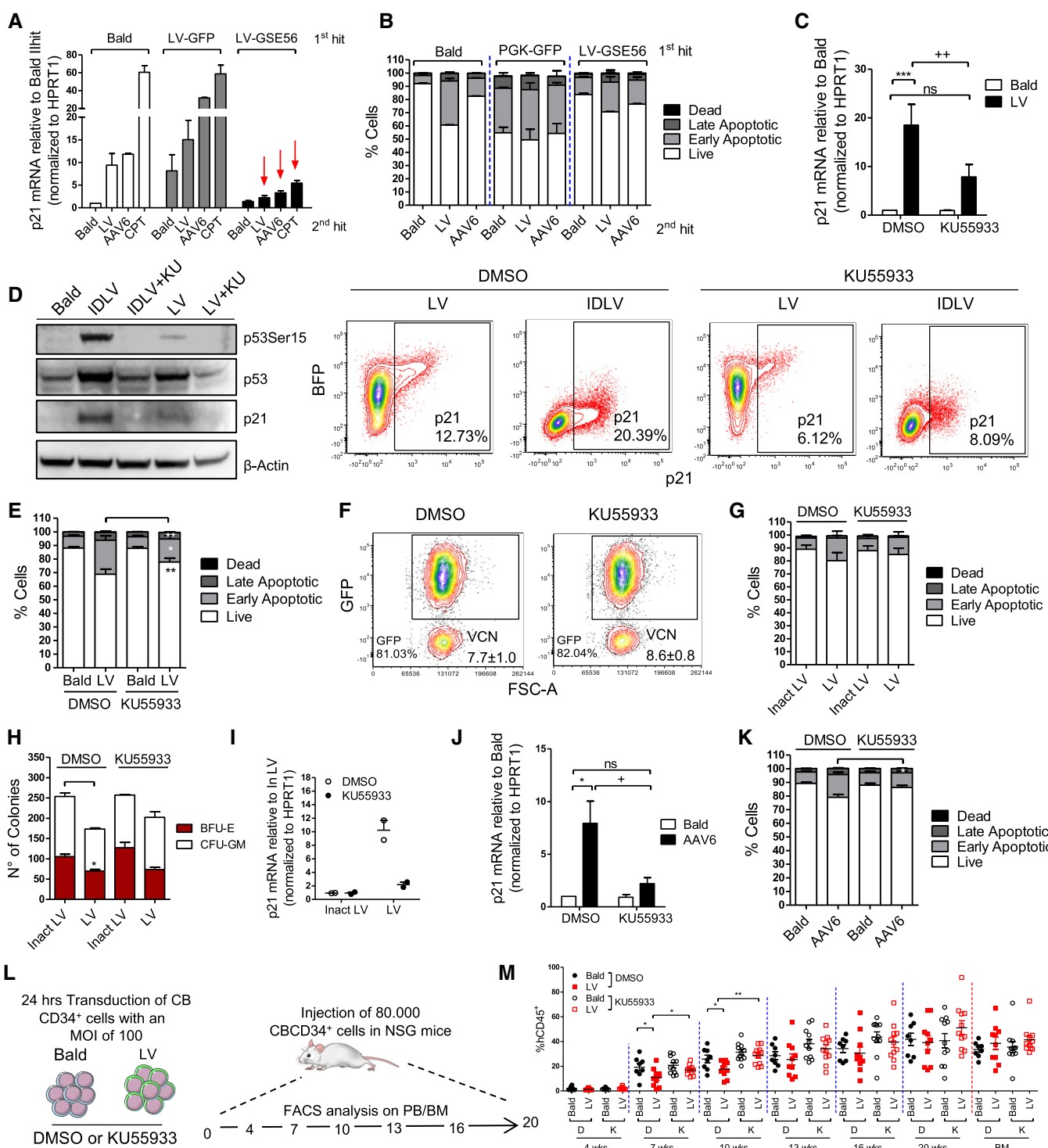

Figure 4.

## Inhibition of p53 activation rescues HSPC apoptosis and engraftment

To test whether blocking the p53 signaling during *ex vivo* HSPC transduction could prevent some of the above-described functional consequences, we first exposed HSPC to a control LV or an LV over-expressing a dominant negative form of p53, GSE56

(Milyavsky *et al*, 2010; Nucera *et al*, 2016), that we verified to efficiently block p53 signaling in HSPC even upon strong DNA damage by camptothecin (CPT) (Fig 4A). Activation of the p53 signaling in terms of p21 induction upon a second round of LV or AAV6 transduction was completely prevented in GSE56-expressing cells compared to control transduced counterparts (Fig 4A). In agreement, reduced apoptosis was observed in

◄

**Figure 4.    Inhibition of LV signaling rescues HSPC apoptosis and engraftment.**

A    p21 mRNA levels measured 24 h after the second hit of transduction and 6 hours after CPT treatment in CB-CD34$^+$ overexpressing a dominant negative form of p53 (GSE56) or GFP as control (mean ± SEM, $n$ = 6 for Bald and LV 2$^{nd}$ hit, $n$ = 2 for AAV6 2$^{nd}$ hit and $n$ = 5 for CPT 2$^{nd}$ hit).

B    Apoptosis was evaluated 48 h after the second hit of transduction with (Bald, LV, or AAV6 vectors) in CB-CD34$^+$ overexpressing GSE56 or GFP as control. (mean ± SEM, $n$ = 2).

C    p21 mRNA induction was measured 48 h post-exposure in CB-CD34$^+$ cells transduced with LV or Bald control at MOI 100 or p24 equivalent in the presence of 10 μM KU55933 (mean ± SEM, $n$ = 9, Dunn's adjusted Kruskal–Wallis test, ***$P$ ≤ 0.001, Wilcoxon signed-rank test, $^{++}P$ = 0.0039).

D    Western Blot ($n$ = 1) (left panel) and representative plot of p21 FACS staining (right panel) were performed on whole mPB-CD34$^+$ and CB-CD34$^+$ 24 and 48 h after the transduction, respectively, in the presence of 10 μM KU55933.

E, F    Apoptosis and transduction efficiency were measured 48 h and 5 days post-exposure, respectively, in CB-CD34$^+$ cells transduced with LV or Bald control at MOI 100 or p24 equivalent in the presence of 10 μM KU55933 (mean ± SEM, $n$ = 15, Wilcoxon signed-rank test, **$P$ = 0.0061, *$P$ = 0.0105, E).

G–I    Impact of ATM inhibition on apoptosis, CFC assay, and p21 mRNA induction on mPB-CD34$^+$ cells after transduction with two hits at MOI 100 of clinical-grade LV or p21 equivalent of inactivated clinical LV (inact LV) as a control (mean ± SEM, $n$ = 3, G), (mean ± SEM, $n$ = 3, Dunn's adjusted Friedman test, *$P$ ≤ 0.05, H) (mean ± SEM, $n$ = 2, I).

J, K    Impact of ATM inhibition on AAV6-mediated induction of p21 mRNA and apoptosis (mean ± SEM, $n$ = 6, Dunn's adjusted Friedman test, *$P$ ≤ 0.05, Wilcoxon signed-rank test, $^+P$ = 0.0313, J), (mean ± SEM, $n$ = 6, Wilcoxon signed-rank test, *$P$ = 0.0313, K).

L, M    In vivo impact of ATM inhibition on total human CD45$^+$ cells in the peripheral blood over time (mean ± SEM, $n$ = 8 for Bald DMSO, $n$ = 10 for LV DMSO, $n$ = 11 for Bald and LV KU55933, Mann–Whitney test, *$P$ = 0.0434,  for Bald vs. LV, *$P$ = 0.0411 for LV D vs. LV K at 7 weeks, **$P$ = 0.0043, M).

Source data are available online for this figure.

GSE56-expressing HSPC upon a second round of transduction (Fig 4B).

p53 signaling can be induced by different upstream signal mediators such as the ATM kinase. Pharmacological inhibition of ATM during LV or IDLV exposure significantly reduced p21 mRNA up-regulation (Fig 4C), decreased p53 phosphorylation and p21 protein levels (Fig 4D), and partially inhibited induction of apoptosis in HSPC (Fig 4E) without compromising transduction efficiencies (Fig 4F). Positive impact of ATM inhibition on apoptosis, colony output, and cell counts in culture together with reduced p21 induction was confirmed also in the more clinically relevant setting of a two-hit transduction protocol in mPB CD34$^+$ cells using a clinical-grade LV (Fig 4G–I). Also AAV-6 mediated p21 induction and apoptosis were prevented by ATM inhibition (Fig 4J–K) while it had no impact of γRV-mediated induction of ISG (Fig EV3L). Of note, although ATM inhibition improved cell survival of transduced HSPC, it did not affect the observed proliferation delay potentially due to residual p21 activity (Fig EV3L). In agreement with lack of p21 induction, no proliferation delay was detected in γRV transduced HSPC (Fig EV3L). Importantly, transient inhibition of p53 signaling during the *ex vivo* transduction procedure improved LV-exposed HSPC engraftment resulting in comparable levels of human CD45$^+$ cells detected in the peripheral blood between transduced and Bald-exposed cells (Fig 4L and M), without altering lineage composition or transduction efficiency in vivo (Fig EV3M and N). Also in this setting long-term human cell engraftment was less affected by LV exposure, further indicating that activation of the p53 signaling upon transduction predominantly impacts short-term HSPC.

These results further confirm that increased apoptosis and lower engraftment observed after LV transduction are dependent on p53 activation in human HSPC and that these responses can be at least partially prevented by transient inhibition of the upstream mediator ATM without affecting transduction efficiency.

## Discussion

Improving cell recovery as well as rapid and robust engraftment of *ex vivo* manipulated HSPC remains a high priority goal for the safe and successful clinical deployment of HSC gene therapy. We have addressed here the impact of LV transduction on the global transcriptional landscape early on upon gene transfer in human HSPC. Very limited changes could be detected with the remarkable absence of innate immune signaling while transduction did activate DNA damage responses. Although HIV-1 has been shown to activate p53 signaling in primary CD4$^+$ T cells as well as in the U2OS cell line (Lau et al, 2005; Cooper et al, 2013), in both cases induction was strictly dependent on viral integration. Furthermore, p21 may not be the preferential downstream effector of p53 activation in this context as we did not observe significant alterations in its expression levels in CD4$^+$ T cells and several other cell lines tested upon exposure to LV. Our findings suggest that activation of the p53 signaling in human HSPC upon LV transduction is DNA break independent as it can be triggered also by non-integrating LV and AAV DNA. Activation of DDR has been shown to occur independently of physical damage to DNA upon chromatin condensation (Burgess et al, 2014). The predominantly quiescent nature of human HSPC could favor break-independent activation of DDR as opposed to the actively proliferating CD4$^+$ T cells or cell lines.

Nuclear sensing of the double-stranded vector DNA triggered ATM-dependent activation of p53 in human HSPC. Phosphorylation of the histone variant H2AX is a key feature of ATM-dependent triggering of a cascade of DDR events, but it is not critical for phosphorylation of ATM substrates such as p53 (Kang et al, 2005). In agreement with a break-independent activation of p53 through ATM, we could not detect significant levels of phosphorylated H2AX foci upon transduction. These results also suggest that HIV-1 integration per se may not robustly recruit the DNA repair machinery in human HSPC, potentially due to steric protection by the viral pre-integration complex (Craigie & Bushman, 2012). Interestingly, aberrant nucleic acid accumulation has been suggested to trigger ATM-dependent DDR responses in mouse embryonic fibroblasts deficient for the cytoplasmic DNA exonuclease Trex1 (Yang et al, 2007). ATM has also been shown to be activated in the presence of free double-strand (ds) DNA ends and short single-stranded (ss) DNA overhangs (Lee & Paull, 2005; Shiotani & Zou, 2009). This type of molecular patterns often characterize viral genetic material and are usually associated with IFN triggering but could also mimic DNA breaks and lead to DDR as reported here.

Both HIV-1 and MLV DNA have been shown to trigger IFN responses through activation of the cytosolic nucleic acid sensor cGAS (Gao *et al*, 2013). There is also emerging evidence regarding cross-talk between innate immune signaling and DDR (Wu *et al*, 2006; Yu *et al*, 2015). Indeed, Poly(I:C)-mediated induction of innate immune responses in HSPC was also accompanied by significant activation of apoptosis-related pathways. Nevertheless, although we carefully searched for evidence of IFN and NF-κB signaling in LV-transduced HSPC, we could not find any significant modulation of these pathways. On the other hand, the MLV-based γRV did trigger substantial expression of several ISG, likely through cytosolic recognition of viral RNA by endosomal TLR or RIG-I, that could be prevented by blocking type I IFN signaling but was not affected by ATM inhibition. Induction of ISG does not *per se* explain why γRV failed to trigger p53, as LV still induced p21 in IFN pre-exposed HSPC. Absence of p53 activation is more likely due to the lower transduction efficiencies of γRV in HSPC. The capacity of both LV and AAV to actively enter into the nucleus of non-dividing, quiescent cells (Bushman *et al*, 2005; Nonnenmacher & Weber, 2012), such as HSPC, could allow them to evade the cytosolic sensors that instead detect the γRV nucleic acids potentially accumulating in the cytoplasm while waiting for mitosis to occur. Nevertheless, removal of the cPPT from the LV did not lead to ISG expression despite reducing significantly p21 mRNA induction. The rates of cytoplasmic accumulation of the ΔcPPT vector may not be sufficient to trigger cytosolic innate sensors. Furthermore, differences in reverse-transcription and/or uncoating rates between LV and γRV or the exploitation of cellular co-factors by HIV-1 could also contribute to avoiding IFN responses to LV in human HSPC (Towers & Noursadeghi, 2014; Sauter & Kirchhoff, 2016). Interestingly, LV transduction did not lead to activation of p53 signaling in murine HSPC, but triggered robust IFN signaling instead. Which are the LV molecular patterns that activate type I IFN in the murine setting remain to be investigated. In human cells, HIV-1 evades innate immune recognition thanks to its capacity to interact with host factors such as cyclophilin A (CypA) and cleavage and polyadenylation specific factor 6 (CPSF6) (Rasaiyaah *et al*, 2013). LV may activate type I IFN responses in murine HSPC due to the absence of some species-specific co-factor interactions. Lack of DDR in this setting could be partly due to the higher proliferation rate of murine HSC as compared to the human counterpart (Doulatov *et al*, 2012) or preceded by the cytoplasmic triggering of type I IFN responses.

Increased p53 activity has been shown to promote HSPC quiescence (Liu *et al*, 2009). We observed that in human HSPC, induction of p53 by LV led to their lower proliferation and a higher fraction of quiescent cells in $G_0$. This, in principle, could preserve the repopulating cells during gene transfer, as quiescence maintains greater stem cell capacity compared with more frequently dividing cells (Wilson *et al*, 2008). Furthermore, the transduced HSPC could engraft more compared to controls *in vivo* due to the LV-induced higher frequency of cells in $G_0$ (Doulatov *et al*, 2012). The parallel induction of apoptosis seems however to counterbalance these potential benefits, in particular in the fraction containing the short-term repopulating HSPC, as significantly lower percentages of human cells were retrieved from mice having received transduced HSPC. On the other hand, decreased p53 levels have been shown to rescue human HSPC from irradiation-induced programmed cell

death (Milyavsky *et al*, 2010) and direct knockdown of p21 has been suggested to improve their engraftment (Zhang *et al*, 2005). We show that inhibition of p53 activation by transiently blocking the upstream activator ATM during LV transduction partially rescued *ex vivo* apoptosis of human HSPC leading to higher engraftment *in vivo*, without affecting gene transfer efficiency. Noteworthy, although ATM inhibition rescued LV-induced apoptosis, it did not impact the reduced HSPC proliferation. These data favor the hypothesis that a window of non-apoptotic quiescence can be reached in these conditions yielding improved HSPC engraftment. In this setting, also control cells seemed to benefit from ATM inhibition suggesting that transplantation *per se* may activate potentially harmful p53 signaling in HSPC, as suggested also by experiments in which p53 knockdown HSPC engrafted more compared to control transduced counterparts even in the absence of irradiation-induced DNA damage (Milyavsky *et al*, 2010).

The transient wave of p53 signaling did not lead to any apparent long-term consequences as engraftment levels tended to normalize over time between untransduced and treated HSPC and the long-term repopulating stem cell frequencies remained unaffected, in agreement with unaltered telomere length and gene expression profiles observed in LV-transduced rhesus macaques HSPC long-term *in vivo* (Sellers *et al*, 2014). Our finding that both integrating and non-integrating vectors do not detectably affect the biological properties of long-term HSPC despite triggering similar molecular responses as observed for the short-term repopulating ones underscores biological differences between these two subsets of HSPC, warranting further investigation in future. The more persistent proliferation arrest observed in the primitive CD34[+]CD133[+]CD90[+] fraction could in part account for better preserving the long-term HSPC engraftment potential. Furthermore, it is possible that the long-term HSPC are less sensitive to DDR, as recently reported for quiescent versus activated murine HSPC (Walter *et al*, 2015).

The negative impact LV-mediated p53 signaling has on the short-term hematopoietic stem and progenitor cell (ST-HSPC) engraftment could be of relevance as rapid engraftment is critical for a safe and successful clinical outcome. Indeed, neutropenia-related infections remain a major cause of mortality and morbidity in both autologous and allogeneic hematopoietic stem cell transplantation (HSCT; Pagano *et al*, 2017; Tomblyn *et al*, 2009) and significant efforts are ongoing to prompt early HSPC engraftment (Lund *et al*, 2015; Kandalla *et al*, 2016; Baron & Nagler, 2017). Prolonged neutropenia due to delayed engraftment remains a major issue also in HSPC gene therapy. Of note, the neutrophil recovery time tends to be slightly longer in this context and does not seem to correlate with the cell dose (Sessa *et al*, 2016). Clonal tracking studies performed in the context of a gene therapy trial to treat the Wiskott–Aldrich syndrome (WAS) suggest that ST-HSPC plays an important role in these first phases of engraftment and hematopoietic reconstitution in humans (Biasco *et al*, 2016). This notion is further supported in the murine setting in which the early phase of hematopoietic reconstitution has been shown to be almost exclusively supported by the ST-HSPC-enriched CD34[+]CD38[+] fraction of HSPC (Zonari *et al*, 2017). Overall, these observations suggest that loss of ST-HSPC may be particularly relevant in settings in which *ex vivo* manipulation of HSPC is required.

Our results suggest that gene therapy vectors can contribute to delayed HSPC engraftment, although we cannot exclude a potential

effect also of the growth conditions during their *ex vivo* manipulation. Moreover, LV-mediated signaling may have more pronounced functional consequences in the context of diseases characterized by an elevated pro-inflammatory state or by genetic defects impacting the DDR pathways. Transient ATM inhibition provides proof-of-principle that dampening this vector signaling could improve hematopoietic reconstitution, although further safety assessments are certainly warranted regarding the potential unwanted effects these approaches may have. Inhibiting p21 induction could also render the second round of LV transduction used in current gene therapy protocols more efficient as p21 has been suggested to restrict HIV-1 integration in human CD34[+] HSPC (Zhang *et al*, 2007). Conversely, induction of DDR pathways by IDLV or AAV donor vectors could potentially benefit targeted HSPC gene editing. Indeed, both IDLV and AAV have been reported to perform efficiently as donor DNA templates, possibly because of the here described induction of DDR and thus repair. On the other hand, also AAV-exposed HSPC have been recently shown to engraft less than their unmanipulated counterparts as shown here for LV-exposed cells (Dever *et al*, 2016). Development of strategies to specifically block the apoptosis-inducing arm of the vector signaling while preserving the component of DDR that potentially promotes HR could further improve targeted gene editing efficiency in human HSPC.

Overall, our studies shed light on the molecular mechanisms and functional consequences of gene therapy vector sensing in human HSPC. Better knowledge regarding these vector–host interactions will allow the development of more stealth gene therapy protocols. This will be of particular relevance in the context of specific disease settings in which vector signaling might impact more dramatically both gene transfer efficiency and HSPC biology. Furthermore, deeper understanding of the signaling cascades activated by non-integrating vector platforms in HSPC will contribute to the design of more efficient therapeutic protocols in the context of the rapidly expanding field of targeted genome editing.

# Materials and Methods

### Viral vectors

Lentiviral vector and γRV were produced by transient transfection in 293T cells and were all VSV-g pseudotyped and concentrated by ultracentrifugation as already described (Montini *et al*, 2006). After transfection and collection, the purified and clinical-grade LV underwent two steps of column chromatography, ion exchange and gel filtration, to purify the vector from contaminants due to transient transfection (DNA and debris). Bald vector, an entry-incompetent LV, was produced omitting the Env-encoding plasmid during vector production. Empty LV, a genome-less lentiviral vector was produced omitting the SIN LV PGK-GFP transfer vector during vector production. To produce the p21 overexpressing LV we replaced the thymidine kinase cDNA of a previously described bidirectional LV expression cassette (Amendola *et al*, 2005), with the cDNA of the human p21 gene, that is express under the control of the human PGK promoter. The GFP is expressed from the minimal CMV promoter in opposite orientation. The GSE56 overexpressing vector is an already described bidirectional vector which expresses GSE56

and GFP in opposite orientation (Nucera *et al*, 2016). Purified LV was kindly provided by Molmed S.p.a (Milan, Italy). Inactivated purified LV was obtaining by heating (1h at 56°C). AAV6-IL2RG-eGFP was kindly provided by Sangamo Biosciences. Self-inactivating VSV-g pseudotyped LV, integrase-defective LV (IDLV), and γ-RV stocks were prepared, concentrated by ultracentrifugation, and titered as previously described (Follenzi *et al*, 2000; Montini *et al*, 2006; Lombardo *et al*, 2007). For clinical-grade LV, a two-step column chromatography, ion exchange followed by gel filtration, was performed to remove contaminants as previously described (Biffi *et al*, 2013). All CD34[+] cells were pre-stimulated with early-acting cytokines 24 h prior to transduction at the indicated multiplicity of infection (MOI) as previously described (Petrillo *et al*, 2015).

### Colony-forming unit (CFU) assay and transplantation of human HSPC in NSG mice

After 10 days, colonies were identified as erythroid or myeloid by morphological criteria, counted, plucked, and pooled into sets of three to obtain DNA for quantification of vector content. (NSG) mice were purchased from Jackson laboratory. Human CB-derived CD34[+] cells were pre-stimulated and transduced as indicated. All animal procedures were performed according to protocols approved by the Animal Care and Use Committee of the Ospedale San Raffaele (IACUC 611) and according to the Italian law. From eight up to 10 weeks of age, NSG mice were sublethally irradiated (radiation dose: 200 cGy for mice weighting 18–25 g and of 220 cGy for mice above 25 g of weight) 24 h prior to xenotransplantation. $8 \times 10^4$ human CB-derived CD34[+] cells were injected into the tail vein of primary NSG mice 24 h after transduction. Peripheral blood was sampled at indicated times post-transplant and analyzed as previously described (Petrillo *et al*, 2015). At sacrifice, the cells from the spleen and BM isolated from the primary recipients were analyzed at flow cytometry and the CD34[+] cells were purified from the BM through positive magnetic bead selection on LD and MS columns (Miltenyi) according to the manufacturer's instructions. Purity was verified by FACS prior to pooling by condition and injection into secondary recipients. Between $9 \times 10^5$ and $1 \times 10^6$ CD34[+] cells isolated from the primary hosts were injected into the tail vein of sublethally irradiated secondary NSG mice (8–10 weeks old). Peripheral blood was sampled at indicated times post-transplant and analyzed as described above. At 14 weeks of age, all mice were sacrificed by $CO_2$ to analyze the BM and the spleen of secondary mice as described above. Cell counts for all *in vivo* experiments before and after treatment, prior to transplantation, are provided in Appendix Table S3.

### Limiting dilution assay

Limiting dilution assay was performed by transplanting into irradiated 8-week-old NSG mice four different doses of CB-CD34[+] cells counted prior to transduction (3,000, 10,000, 30,000, or 90,000) either for Bald and LV groups. Cells were injected 24 h after the transduction. A mouse was scored as positively engrafted if it had > 0.1% engraftment in multiple lineages in the BM at the time of sacrifice (16 weeks). HSC frequency was estimated by linear regression analysis and Poisson statistics using publicly available ELDA

## The paper explained

### Problem

Clinical application of lentiviral vector (LV)-based hematopoietic stem and progenitor cell (HSPC) gene therapy is rapidly becoming a reality. Nevertheless, LV-mediated signaling and its potential functional consequences on HSPC biology remain poorly understood. Given the viral origin of LV, exposure of human HSPC to the vector during *ex vivo* gene transfer can trigger signaling events that mimic host cell responses to viral infection with potential short and long-term consequences that have not been addressed to date.

### Results

We have investigated here how lentiviral transduction alters the global transcriptional landscape of human HSPC. Remarkably, LV transduction has a very limited impact in HSPC and the vector completely avoids all innate immune recognition, as opposed to gammaretroviral vectors that robustly activate antiviral responses in HSPC. Nevertheless, LV transduction was not completely stealth. In fact, p53 signaling and DNA damage responses were significantly activated upon exposure of HSPC to both laboratory-grade and clinical-grade LV. We demonstrate that this induction occurs through ATM-dependent recognition of the reverse-transcribed LV DNA intermediates prior to integration. These responses can be triggered also by non-integrating Adeno-associated vectors (AAV), which are used similarly to integrase-defective LV (IDLV) to transiently transduce HSPC for instance in the context of gene editing applications. Given the pivotal role of the p53 signaling pathway in HSPC biology, we addressed the potential functional consequences of LV-mediated p53 activation and show that it results in delayed HSPC proliferation, cell cycle arrest and slightly but significantly increased apoptosis in culture. As a consequence, reduced engraftment of transduced cells was observed *in vivo*. Importantly, transient pharmacological inhibition of the LV-induced signaling rescued both apoptosis and engraftment of HSPC, providing proof-of-principle of a novel strategy to further improve the safety and efficacy of LV-mediated *ex vivo* HSPC gene therapy.

### Impact

Our work provides insight into the signaling and functional consequences of vector platforms relevant for the rapidly expanding fields of HSPC gene therapy and editing, also providing proof-of-principle on how to render these approaches more stealth and safe.

(extreme limiting dilution analysis, http://bioinf.wehi.edu.au/software/elda/) software (Hu and Smyth, 2009).

### Homing assay

For homing assays, $5 \times 10^5$ CB-derived CD34$^+$ cells (pre-treatment dose) were transplanted 24 h after transduction into irradiated 8-week-old NSG mice that were sacrificed 16 h post-transplantation for analysis. The whole bone marrow was harvested from the lower leg of the mice. During the FACS analysis, cell count beads (Flow-Count Fluorospheres) by BECKMAN COULTER were added in each sample to estimate the absolute number of human CD34$^+$ cells per sample. Cell counts for all *in vivo* experiments before and after treatment, prior to transplantation, are provided in Appendix Table S3.

### RNA-Seq

Sequencing was performed on the Illumina HiSeq 2000 platform using SBS 2x100PE protocol (Smyth, 2004). Pathway analysis was initially performed using EnrichrR platform (Chen *et al*, 2013); most of the advanced network modeling was performed using Cytoscape. The complete RNA-Seq dataset is available at NCBI, accession number GSE92652.

### Flow cytometry

FACS on HSPC was performed as previously described (Petrillo *et al*, 2015). The apoptosis assays were performed with the Annexin V Apoptosis Detection Kit I (BD Pharmingen) according to the manufacturer's instructions. The cell cycle analysis was performed by Ki67 (BD Pharmingen) and Hoechst (Invitrogen) staining as previously described (Lechman *et al*, 2012).

### Statistical analysis

Data are expressed as mean ± SEM. Nonparametric Wilcoxon signed-rank test was used to assess the fold of expression of specific genes respect to the internal control set value as 1. Comparison of multiple groups was performed with nonparametric (Kruskal–Wallis) for unpaired dataset, while the nonparametric (Friedman test) was used for matched paired observations. In both cases, Dunn's adjustment was used for multiple comparisons. Nonparametric unpaired Mann–Whitney test was used to assess the difference between two groups. Significance was considered at $P < 0.05$. The number of samples analyzed and the statistical test used are indicated in the legends for each figure.

For further details see the Appendix Supplementary Methods.

**Expanded View** for this article is available online.

### Acknowledgments

This work was supported by the Italian Ministry of Health, the Fanconi Anemia Research Foundation, and the Italian Telethon Foundation to AKR; EU and Italian Telethon Foundation grants to LN. We thank Cesare Covino from Alembic for help with the confocal imaging acquisition and analysis; Alessandro Aiuti, Alessandra Biffi, and MolMed Spa for the clinical-grade LV used in this study. FP and CP conducted this study as partial fulfillment of their PhD in Molecular Medicine, Program in Cellular and Molecular Biology, International PhD School, Vita-Salute San Raffaele University, Milan, Italy.

### Author contributions

FP and CP conducted experiments and analyzed data. FP prepared the RNA-Seq libraries, DL performed RNA-Seq runs and quality control. ES helped design the RNA-Seq experiments. MR and DC performed bioinformatic and statistical analysis of NGS data. IC and SB provided technical. BG provided reagents and intellectual input. FP, DC, LN, and AK-R designed the research studies, analyzed data, and wrote the manuscript.

### Conflict of interest

AK, FP, BG, and LN are inventors on pending and issued patents on lentiviral vector technology and gene transfer filed by the Salk Institute, Cell Genesys, or Telethon Foundation and San Raffaele Scientific Institute.

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
