## [Review Process File · EMBO Molecular Medicine]

Lentiviral Vectors Escape Innate Sensing but Trigger p53 In Human Hematopoietic Stem and Progenitor Cells

Francesco Piras, Michela Riba, Carolina Petrillo, Dejan Lazarevic, Ivan Cuccovillo, Sara Bartolaccini, Elia Stupka, Bernhard Gentner, Davide Cittaro, Luigi Naldini and Anna Kajaste-Rudnitski

Corresponding author: Anna Kajaste-Rudnitski, San Raffaele Scientific Institute

Review timeline:

Submission date:	21 April 2017
Editorial Decision:	17 May 2017
Revision received:	30 May 2017
Editorial Decision:	02 June 2017
Revision received:	06 June 2017
Accepted:	07 June 2017

Transaction Report:

(Note: Please note that the manuscript was previously reviewed at another journal and the reports were taken into account in the decision making process at EMBO Molecular Medicine. Since the original reviews are not subject to EMBO's transparent review process policy, the reports and author response cannot be published.)

Editor: Roberto Buccione

1st Editorial Decision

17 May 2017

Thank you for the submission of your manuscript and previous review correspondence to EMBO Molecular Medicine.

We have now heard back for the expert advisor who was asked to evaluate it. As you will see, s/he is globally positive but points to a few issues that need further clarification. You will also note that, not surprisingly in my opinion, a few reservations are expressed concerning the degree of clinical impact, that call for some deemphasizing.

I am prepared to make an editorial decision on the next, final version of your manuscript, provided you carefully and fully address the advisor's concerns. Please highlight the changes in the manuscript text.

Please submit your revised manuscript within two weeks. I look forward to seeing a revised form of your manuscript as soon as possible.

***** Reviewer's comments *****

Referee #1 (Remarks):

- Well performed study
- This is novel since reporting consequences (innate immune-DDR responses) of LV (γ RV, AAV)

transduction of human hematopoietic progenitor cells had not yet been done. It is a significant advance in describing the biology underlining gene therapy based on viral vectors

- Potential to increase early stages of hematopoietic reconstitution following LV mediated GT is a little but overemphasized since :
 - a) neutropenia related death as mentioned p19, following SCTs is no longer as high as reported in the cited 2009 paper
 - b) it is not excluded that p53 inhibition or ATM inhibition might exert unwanted effects not seen in the experimental setting. I therefore suggest to temper the presentation of the medical interest of this approach
- P3, DAI (NLR family) could also be cited as a cytosolic receptor of DNA
- All over the manuscript, the term "HSC" is used to define cord blood or bone marrow CD34+ cells. This is not fully accurate since only small fractions are bonafide HSC. The term of hematopoietic progenitor cells would thus be more appropriate.

1st Revision - authors' response

30 May 2017

We thank the reviewers for retaining our work technically well performed, novel and of significant advance in understanding the biology underlying gene therapy based on viral vectors. We have modified the manuscript based on the reviewers' insightful criticisms and suggestions and believe to have properly addressed most of his/her concerns and to have significantly improved the overall quality of this work.

Briefly, we have now modified the manuscript to address the concerns specifically raised by the Reviewer as follows:

Referee #1 (Remarks):

- Potential to increase early stages of hematopoietic reconstitution following LV mediated GT is a little but overemphasized since :
 - a) neutropenia related death as mentioned p19, following SCTs is no longer as high as reported in the cited 2009 paper

Regarding this specific point, we thank the Reviewer for this observation and have now modified the discussion on neutropenia-related mortality in hematopoietic stem cell transplantation (HSCT) accordingly and have included more recent references, page 18 of the revised manuscript, as suggested by the Reviewer.

- b) it is not excluded that p53 inhibition or ATM inhibition might exert unwanted effects not seen in the experimental setting. I therefore suggest to temper the presentation of the medical interest of this approach

As the Reviewer correctly suggests, there are certainly some safety concerns associated with the potential application of transient ATM inhibition during ex vivo HSPC gene therapy to be taken into account and carefully addressed before any clinical implementation can be foreseen. Based on this useful comment, we have now tempered the discussion regarding the applicability of ATM inhibition, page 19 of the revised manuscript, acknowledging these relevant safety concerns, as rightly pointed out by the Reviewer.

- P3, DAI (NLR family) could also be cited as a cytosolic receptor of DNA

As the Reviewer rightly points out, also other cytosolic nucleic acid sensor may be involved in vector sensing in HSPC. In particular, we have now cited also the DNA-dependent activator of interferon-regulatory factors (DAI) as a potential sensor of vector nucleic acids in the introduction, page 3 of the revised manuscript, as suggested by the Reviewer

- All over the manuscript, the term "HSC" is used to define cord blood or bone marrow CD34+ cells. This is not fully accurate since only small fractions are bonafide HSC. The term of hematopoietic progenitor cells would thus be more appropriate.

We fully agree with the Reviewer that including the progenitor compartment in our definition of CD34⁺ population is more appropriate. Accordingly, we have modified the term “HSC” to “HSPC” throughout the revised version of manuscript in order to refer more correctly to the heterogeneous population of CD34⁺ hematopoietic stem and progenitor cells, as kindly suggested by the Reviewer.

Finally, we have also modified our manuscript to comply to the EMBO Molecular Medicine editorial requirements as follows:

- We have included five keywords, page 1 of the revised manuscript;
- We have moved the methods and related references of Viral Vectors Colony-forming unit (CFU) assay and Transplantation of human HSPC in NSG mice, Statistical analysis, Limiting Dilution Assay and Homing Assay from the supplementary materials to the main manuscript as requested, pages 21-24;
- We have updated the references to comply the EMBO Molecular Medicine guidelines;
- We have updated all figure legends to comply to the Author Guidelines regarding statistical testing, pages 32-36;
- We have filled-in the complete Author Checklist;
- We have included source data for all the Western blots shown in the main and supplementary figures of the manuscript;
- We have included a statement in the Materials and Methods section of the revised manuscript identifying the institutional and/or licensing committee approving the experiments, page 22;
- We have included a Synopsis of our findings and prepared a visual abstract accompanying it;
- We have included the ORCID ID for the corresponding author.

2nd Editorial Decision

02 June 2017

Thank you for the submission of your revised manuscript to EMBO Molecular Medicine.

I am pleased to inform you that we will be able to accept your manuscript pending a few final amendments concerning items that were not fully dealt with:

I look forward to seeing a revised form of your manuscript as soon as possible.

2nd Revision - authors' response

06 June 2017

Authors made requested editorial changes.

Corresponding Author Name: Anna Kajaste-Rudnitski

Manuscript Number: EMM-2017-07922